# Research

microbiology, immunology, genetics

honeybees, microbiome, innate immunity, symbiosis, colonization resistance, immune priming

**Authors for correspondence:**
Richard D. Horak
e-mail: rdhorak1@gmail.com
Nancy A. Moran
e-mail: nancy.moran@austin.utexas.edu

# Symbionts shape host innate immunity in honeybees

Richard D. Horak, Sean P. Leonard and Nancy A. Moran

Department of Integrative Biology, The University of Texas at Austin, Austin, TX 78712, USA

 RDH, 0000-0003-0630-5481; NAM, 0000-0003-2983-9769

The gut microbiome plays a critical role in the health of many animals. Honeybees are no exception, as they host a core microbiome that affects their nutrition and immune function. However, the relationship between the honeybee immune system and its gut symbionts is poorly understood. Here, we explore how the beneficial symbiont *Snodgrassella alvi* affects honeybee immune gene expression. We show that both live and heat-killed *S. alvi* protect honeybees from the opportunistic pathogen *Serratia marcescens* and lead to the expression of host antimicrobial peptides. Honeybee immune genes respond differently to live *S. alvi* compared to heat-killed *S. alvi*, the latter causing a more extensive immune expression response. We show a preference for Toll pathway upregulation over the Imd pathway in the presence of both live and heat-killed *S. alvi*. Finally, we find that live *S. alvi* aids in clearance of *S. marcescens* from the honeybee gut, supporting a potential role for the symbiont in colonization resistance. Our results show that colonization by the beneficial symbiont *S. alvi* triggers a replicable honeybee immune response. These responses may benefit the host and the symbiont, by helping to regulate gut microbial members and preventing overgrowth or invasion by opportunists.

## 1. Introduction

Honeybees (*Apis mellifera*) harbour a distinctive gut microbiome that is key to their health [1]. Having co-evolved with social bees for over 80 million years, 95% of gut-dwelling organisms fall within nine species clusters of host-specific bacteria spatially organized within bee hindgut compartments [2–4]. The honeybee gut provides an enticing model system for studying host–microbe interactions and understanding the mechanisms by which gut bacteria influence their hosts [5].

*Snodgrassella alvi* colonizes the honeybee ileum and grows in contact with the gut epithelia. Cells of another core gut species, *Gilliamella apicola*, grow on top of this *S. alvi* layer to form a dense biofilm [2,6]. Honeybees with perturbed gut communities die at higher rates when challenged by the opportunistic pathogen *Serratia marcescens* [7]. Thus, the bee gut community protects the host from infection. This protection, or 'colonization resistance', is one of the most widespread benefits provided by symbiotic communities to hosts, including mammalian hosts [8,9]. However, the mechanism of this protection in bees is unclear. Microbiome-derived protection may be caused by active symbiont colonization that physically blocks or antagonizes pathogens.

The dense biofilm formed by *S. alvi* could block pathogen access to host epithelial cells and sequester nutrients [6]. *S. alvi* may directly antagonize invaders with its type VI secretion system and diverse array of effectors [10]. Additionally, metabolism by bee gut microbial members lowers gut lumen pH and oxygen levels [11], and produces short-chain fatty acids (SCFAs) which can inhibit pathogen virulence and growth in mice [12]. Finally, immune priming may also be responsible, a process by which bacteria activate the host innate immune system, making the host more resistant to subsequent bacterial encounters [1,12].

Insect antibacterial immunity relies heavily on the Toll and Imd pathways of the innate immune system [13,14]. These pathways are best-studied in *Drosophila melanogaster,* in which Toll receptors and peptidoglycan recognition proteins (PGRPs) react to bacterial motifs [13]. For Toll, binding of the endogenous ligand Spaetzle with the Toll receptor triggers the degradation of Cactus, an inhibitor of the NF-kB transcription factor Dorsal [15]. The Imd pathway is triggered by direct binding of peptidoglycan with PGRPs, signalling cleavage of a self-inhibitory region of the NF-kB-like transcription factor Relish by the protein Dredd [16]. Dorsal and Relish then translocate into the host nucleus and upregulate immune effectors, including antimicrobial peptides (AMPs)—short peptides that perforate bacterial membranes and inhibit protein folding [14,17]. Honeybees encode orthologues for all core proteins of these pathways as well as a unique ensemble of AMPs: *abaecin, apidaecin, defensin* and *hymenoptaecin* [18–22].

Honeybees upregulate expression of *apidaecin* in response to *S. alvi* colonization [23,24]. Combined with previous data supporting symbiont-based pathogen protection, these results support a role for *S. alvi* in honeybee immune priming. However, neither study measured transcriptional responses of fat bodies—the centre of the insect immune response [14]. Additionally, previous studies detailing this protective effect used only live bacterial inoculations, so they were unable to separate the impacts of colonization and immune priming.

In this study, we investigated the role *S. alvi* plays in activating the honeybee immune system and protecting hosts from pathogens. Using both live and heat-killed bacteria, we show that colonization is not required for pathogen protection, but aids in pathogen clearance, and that heat-killed *S. alvi* causes a more extensive immune expression response than the live symbiont. These results suggest that *S. alvi* is not only priming the honeybee immune system but also potentially modulating it.

## 2. Methods

### (a) Bacterial strains
*Escherichia coli* strain MG1655 was grown at 37°C on LB agar. *S. alvi* strain wkB2 and *S. marcescens* strain N10 were grown as previously described, on Columbia agar supplemented with sterile sheep blood at 35°C and 5% $CO_2$ [2]. *S. alvi* and *S. marcescens* isolates were previously derived from honeybee guts.

### (b) Preparation of heat-killed cells
Overnight growths of *E. coli* and *S. alvi* on agar plates were pooled individually and suspended in 500 μl PBS. The $OD_{600}$ of bacterial cultures was measured, and the amount of *S. alvi* and *E. coli* used for heat-killing was diluted to an OD representative of $5 \times 10^8$ CFUs. Heat-killed cells were prepared by then suspending *S. alvi* or *E. coli* in 1 ml of PBS and heating to 80°C. Cells were left at 80°C for 30 min for *S. alvi* and 10 min for *E. coli*. Direct plating of heat-killed bacterial suspensions confirmed that no live cells were present.

### (c) Honeybee collection and containment
All *Apis mellifera* samples were obtained from hives at the University of Texas at Austin. To obtain microbiota-free bees, dark-eyed pupae were removed from capped brood cells using sterilized forceps and placed in sterile plastic cages as previously described [25]. Bees obtained this way are 'microbiota-free' and lack the usual gut bacterial species [26]. Newly emerged bees were fed sterile pollen and 1 : 1 filter-sterilized sucrose water until fully matured 3 days later. Bees were randomly assigned to groups of approximately 20 adult bees and then chilled at 4°C and placed in 50 ml centrifuge tubes corresponding to their treatment group. Bee groups were inoculated by feeding with 1 ml filter-sterilized sucrose water (in a 1 : 1 mix of sucrose and water) or 800 μl filter-sterilized sucrose water and 200 μl of their bacterial or heat-killed bacterial treatment. Bees were then transferred to plastic cup cages and fed sterile pollen. Each cage was given 10 ml of 1 : 1 filter-sterilized sucrose water mixed with their respective treatment.

### (d) Gene expression analysis
Bees were sampled on their respective days post bacterial inoculation and moved into 15 ml centrifuge tubes to be frozen at −80°C. Abdomens were removed from each bee and placed into 600 μl of RNA lysis buffer and pestle homogenized. RNA was then isolated using a Zymo Research Quick-RNA Tissue/ Insect Microprep Kit (catalogue no. R2030). RNA was eluted in 50 μl of RNase-free water and stored at −80°C. Concentrations were quantified using a Thermo Fisher Scientific NanoDrop Lite Spectrophotometer. RNA was reverse-transcribed using a Quanta-Bio qScript cDNA Synthesis Kit (catalogue no. 95047-500) with all reactions normalized to 500 ng μl$^{-1}$ of input RNA. Relative gene expression was determined using quantitative PCR. The source of primers used in qPCR assays can be found in electronic supplementary material, table S1. Absolute quantification of the housekeeping gene RPS18 was accomplished by cloning RPS18 into the Promega pGEM-T vector (catalogue no. A1360). Primer efficiency was determined using 10-fold serial dilutions of primer standards. Standards for target genes not cloned into a pGEM-T vector were created by PCR amplification of the control group cDNA with target gene qPCR primers. Reactions were run in an Eppendorf MasterCycler Realplex machine using technical replicates for each sample and Bio-Rad iTaq Universal SYBR Green Supermix (catalogue no. 1725121) for fluorescence. Relative gene expression for the genes, *abaecin, apidaecin, hymenoptaecin, cactus-1, cactus-2, dorsal, relish, dredd, pgrp-lc, pirk* and *toll*, were determined using a 2-ΔΔCt method and log$_2$ transformed [27].

### (e) Survival and clearance assays
Microbiota-free bees were fed their respective bacterial, heat-killed bacterial or PBS treatment. Five days post-inoculation bees were chilled and placed in 0.5 ml microcentrifuge tubes, each with a hole cut in the bottom. Each bee was then hand-fed 5 μl of a mixture of 20% sucrose and 80% PBS or this same mixture plus a 24 h growth of *S. marcescens* diluted to an $OD_{600}$ of 1 for survival assays and 0.5 for clearance assays. In survival assays, bees were then returned to their respective cup cages and the number of dead bees was recorded daily for 10 days. Survival data were pooled from three independent trials. A Cox proportional hazard model was used to test any effect independent trials might have on bee survivorship (electronic supplementary material, figure S1). In the clearance assay, whole guts were extracted from bees 1 day and 3 days post-inoculation with *S. marcescens.* These guts were immediately homogenized into 200 μl PBS. We prepared serial dilutions and spot-plated 10 μl of each dilution onto LB agar plates. *S. marcescens* colonies were counted after 1 day at 37°C in aerobic conditions and total CFUs per gut calculated. Less than 10 discernible CFUs were found in bees given no bacterial exposure.

### (f) Statistical analysis
Survival curves were created using the survminer and survival packages in RStudio (https://rstudio.com/) [28,29]. Curves are based on a Kaplan–Meier fit [30]. The Forrest plot was based on a Cox proportional hazard model [31]. Statistical analysis was

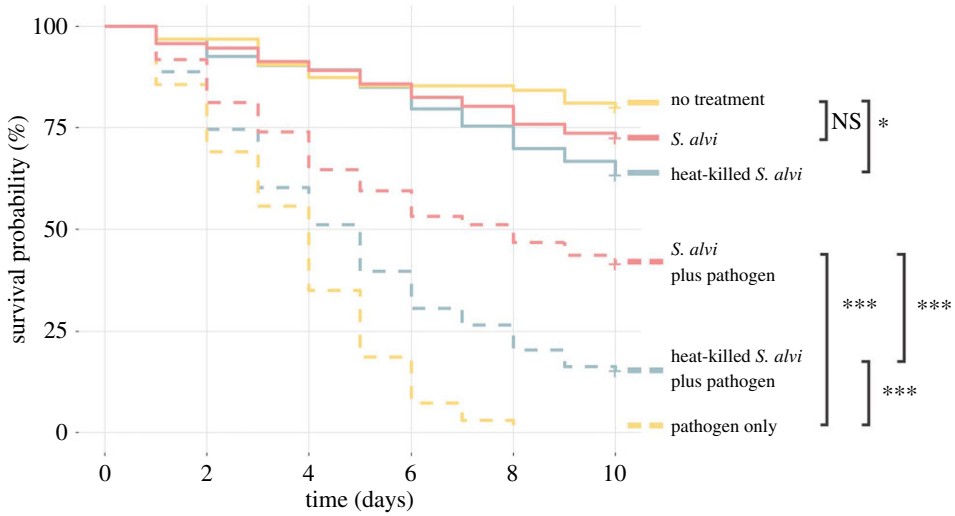

**Figure 1.** Inoculation with live and heat-killed *S. alvi* provides pathogen protection. Survival of bees from different treatment groups after inoculation with PBS or 5 µl of the pathogen *S. marcescens* at $OD_{600} = 1$. When challenged with *S. marcescens*, bees previously treated with live *S. alvi* or heat-killed *S. alvi* survived better than bees given no treatment prior to pathogen exposure. *p*-values obtained using a log-rank test with Benjamini–Hochberg correction. Total N = 570 bees across three replicate experiments. *** = $p < 0.001$, ** = $p < 0.01$, * = $p < 0.05$, NS = not significant. (Online version in colour.)

accomplished using a pairwise log-rank test with a Benjamini–Hochberg correction for multiple testing [32,33]. Gene expression graphs were created using the ggplot2 library in RStudio [34]. Statistical analysis for gene expression assays and *S. marcescens* clearance assays was accomplished using the Tukey honest significant difference method [35].

## 3. Results

### (a) Heat-killed *Snodgrassella alvi* protects against the pathogen *Serratia marcescens*

Honeybees containing core microbiome members have higher survival following challenge with the opportunistic pathogen *S. marcescens* [7]. This protection may happen by physically blocking pathogen colonization, actively killing or suppressing pathogens, or priming the host immune system. Therefore, to determine if symbiont colonization is required for protection, microbiota-free honeybees were treated with live *S. alvi*, heat-killed *S. alvi* or sterile sugar syrup. Five days post-treatment, each bee was hand-fed *S. marcescens* or sterile PBS. Across treatments, bees receiving no pathogen had higher survival than bees challenged with the pathogen (figure 1). Bees inoculated with live *S. alvi* showed higher survival compared to those lacking *S. alvi* when challenged with *S. marcescens* (figure 1). Despite lacking the ability to colonize the gut, heat-killed *S. alvi* still provided significant protection compared to bees not exposed to any form of *S. alvi*. But this protection was less than that provided by live *S. alvi* (figure 1). Additionally, bees inoculated with heat-killed *S. alvi* experienced lower survival, suggesting a possible adverse effect to this treatment (figure 1). These trends are consistent across individual experiments (electronic supplementary material, figure S1). Thus, colonization is not required for improved survival after pathogen challenge.

### (b) Both live and heat-killed *Snodgrassella alvi* trigger upregulation of antimicrobial peptides

Protection against *S. marcescens* by heat-killed *S. alvi* supports immune priming as a factor in pathogen defence. To determine

if live and dead *S. alvi* trigger a host immune response, we treated microbiota-free bees with live *S. alvi*, heat-killed *S. alvi* and heat-killed *E. coli* and assessed expression of immune genes 5 days later. While *S. alvi* is a co-evolved symbiont of honeybees, *E. coli* is not a typical bee gut microbiome member. However, both bacteria are Gram-negative. Therefore, by including heat-killed *E. coli* we could assess whether gene expression responses were general to Gram-negative bacterial components or if they were symbiont-specific. In all treatments, we found significant upregulation of the AMPs *abaecin*, *apidaecin* and *hymenoptaecin* relative to uninoculated microbiota-free bees (figure 2a). In a replicated experiment, sampling bees 1, 2 and 5 days post-inoculation, we saw similar upregulation, suggesting that this response spans sampling dates (electronic supplementary material, figure S2A). These results show that both live and dead *S. alvi* trigger an immune response and that this response is not symbiont-specific, as heat-killed *E. coli* treatments show similar upregulation of AMPs.

### (c) Heat-killed *Snodgrassella alvi* trigger a more extensive immune expression response than live *Snodgrassella alvi*

In *Drosophila*, Dorsal and Relish play critical roles as transcription factors in the expression of AMPs [14,17]. To further explore the upregulation of immune effectors following exposure to live or dead *S. alvi*, we measured relative gene expression via qPCR of *dorsal*, *relish* and *cactus* after inoculation with live *S. alvi*, heat-killed *S. alvi* or heat-killed *E. coli*. Surprisingly, bees inoculated with live *S. alvi* did not have as high upregulation compared to both heat-killed treatments (Figure 2b). In a replicated experiment, sampling bees 1, 2 and 5 days post-inoculation, we saw similar patterns (electronic supplementary material, figure S2B). There appears to be no correlation between transcriptional regulator gene expression and AMP gene expression, as all treatment groups had similarly elevated AMP expression regardless of *dorsal* or *relish* expression (figure 2a,b). Interestingly, while *S. alvi* contains components sufficient for immune regulator upregulation,

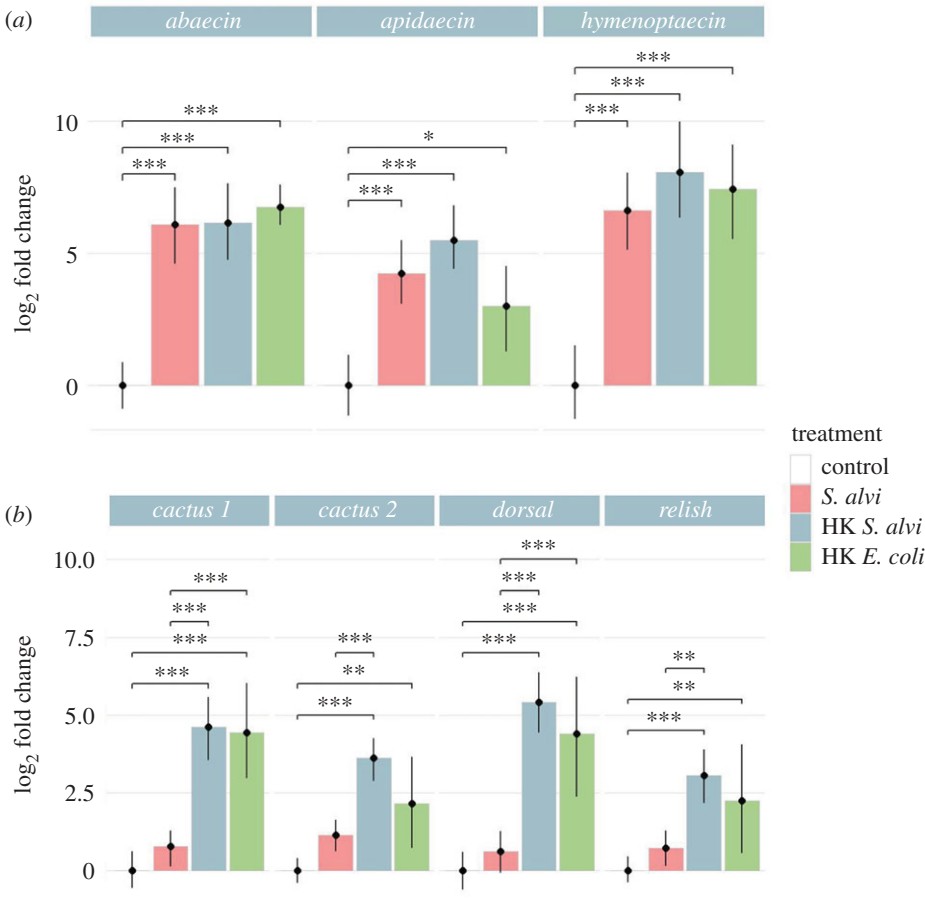

**Figure 2.** Live and heat-killed *S. alvi* trigger differential host immune gene expression. Bee gene expression relative to the housekeeping gene *RPS18* measured from bee whole abdomens 5 days post-treatment. HK = heat-killed. (*a*) Bees inoculated with live *S. alvi*, heat-killed *S. alvi*, and heat-killed *E. coli* trigger upregulation of AMPs compared to the uninoculated control group. (*b*) Bees inoculated with heat-killed *S. alvi* or *E. coli* had significantly higher expression of immune regulatory genes than the control or live *S. alvi* groups. Total N = 73 bees from one hive. *** = $p < 0.001$, ** = $p < 0.01$, * = $p < 0.05$. *p*-values found using Tukey honest significant difference method. (Online version in colour.)

live cells do not trigger the same magnitude of response found in heat-killed treated groups.

### (d) A mixture of live and heat-killed *Snodgrassella alvi* lowers the expression of Imd pathway genes while upregulating toll pathway components

To further discern the impact of *S. alvi* on bee immune gene expression, we expanded our gene targets to include the receptor *toll* and *pgrp-lc*, as well as the Imd regulatory genes *pirk* and *dredd*. In *Drosophila* sp. Pirk is known to inhibit the function of the Imd pathway while Dredd modifies Relish, triggering its translocation to the nucleus [16,36]. Additionally, we added a treatment containing an equal mixture of live and heat-killed *S. alvi*. If live *S. alvi* alone fails to upregulate immune genes, we would expect signals generated by heat-killed *S. alvi* to dominate. However, if live *S. alvi* actively suppresses host immune gene upregulation, we would expect a gene expression pattern more comparable to live *S. alvi* alone. While live and heat-killed *S. alvi* treatments on their own confirmed previous results, our mixed treatment did not favour one or the other (figure 3).

We saw considerable upregulation of AMPs *apidaecin* and *hymenoptaecin* but lower levels of *abaecin* in the mixed treatment group (figure 3*a*). For the Toll pathway genes *toll*, *cactus* and *dorsal*, the mixed treatment group favoured expression similar to treatment with heat-killed *S. alvi* alone (figure 3*b*). While we

found no change in *pirk* levels, the expression of the Imd pathway genes *dredd*, *pgrp-lc*, and *relish* in bees treated with both live and heat-killed *S. alvi* was reduced compared to live or heat-killed treatments alone (figure 3*c*). These results show a context-dependent loss of Imd pathway gene expression when bees were treated with a mixture of *S. alvi* and heat-killed *S. alvi*. Additionally, these results suggest that live *S. alvi* can reduce the expression of Imd pathway genes activated by heat-killed *S. alvi*.

### (e) Live *Snodgrassella alvi* aids in the clearance of the pathogen *S. marcescens* while heat-killed *Snodgrassella alvi* does not

Both live and heat-killed *S. alvi* increase host survival following subsequent challenge with *S. marcescens*, and both trigger differential immune gene expression. To determine if these treatments could aid in *S. marcescens* clearance, microbiota-free bees were inoculated with live *S. alvi*, heat-killed *S. alvi* or a mixture of the two, then challenged with *S. marcescens*. Treatment with live *S. alvi* reduced *S. marcescens* CFUs both 1 day and 3 days after pathogen challenge (figure 4). Bees treated with a mixture of live and heat-killed *S. alvi* showed a similar reduction (figure 4). However, bees fed only heat-killed *S. alvi* showed no significant reduction in pathogen CFUs (figure 4). These findings indicate that colonization by live *S. alvi* boosts

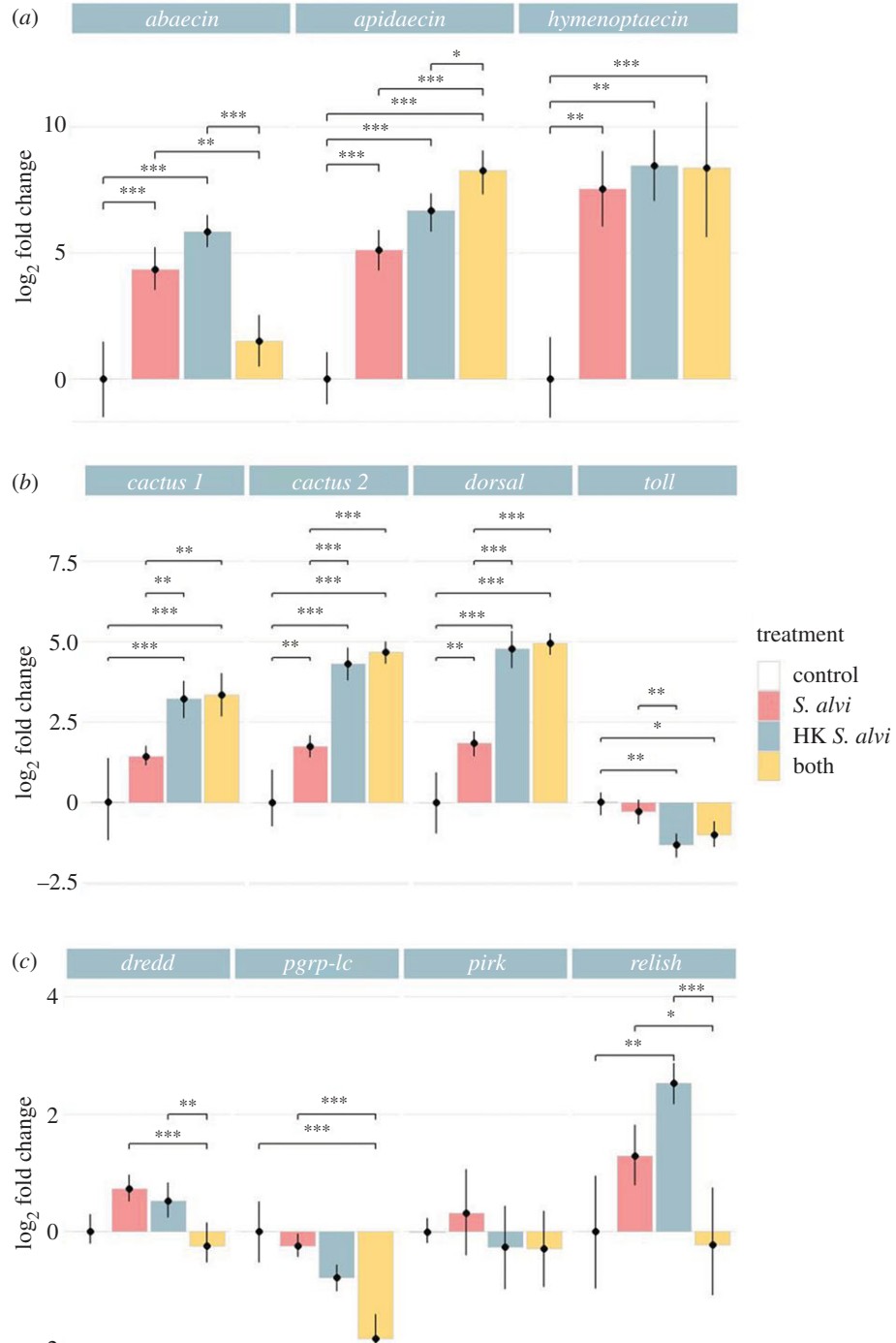

**Figure 3.** Loss of host Imd pathway gene expression post-inoculation with both live and heat-killed *S. alvi*. Bee gene expression relative to the housekeeping gene *RPS18* measured using qPCR from cDNA derived from bee whole abdomens 5 days post-treatment. HK = heat-killed. (*a*) Bees treated with live or heat-killed *S. alvi* trigger AMP upregulation. Mixed treatment of both live and heat-killed *S. alvi* triggers upregulation of *apidaecin* and *hymenoptaecin*, but not *abaecin*, compared to the control group. (*b*) Bees treated with live *S. alvi*, heat-killed *S. alvi*, or both show upregulation of Toll pathway genes *cactus* and *dorsal* compared to the control group. Bees treated with heat-killed *S. alvi* or both live and heat-killed *S. alvi* trigger downregulation of the gene *toll*. (*c*) Bees treated with both live and heat-killed *S. alvi* show lower expression of Imd pathway genes *dredd*, *pgrp-lc*, and *relish* compared to other groups. Total N = 48 bees from one hive. *** = $p < 0.001$, ** = $p < 0.01$, * = $p < 0.05$. *p*-values obtained using Tukey honest significant difference method. (Online version in colour.)

pathogen resistance, increasing bee survival and suppressing pathogen proliferation. However, pre-treatment with heat killed *S. alvi* supports tolerance, increasing bee survival but failing to reduce pathogen numbers.

## 4. Discussion

Increased survival after pathogen challenge in honeybees treated with heat-killed *S. alvi* suggests that immune priming

underlies at least part of this symbiont's protective effect (figure 1). Inoculation with live *S. alvi*, however, leads to higher survival after pathogen challenge, similar AMP upregulation, and increased pathogen clearance. These results together suggest that *S. alvi*'s colonization of the gut ileum's epithelial wall plays a key role in colonization resistance (figures 1, 2*a* and 4). These results reinforce earlier findings that *S. alvi* induces the upregulation of immune genes [23,24]. We observed greater upregulation of AMP genes than

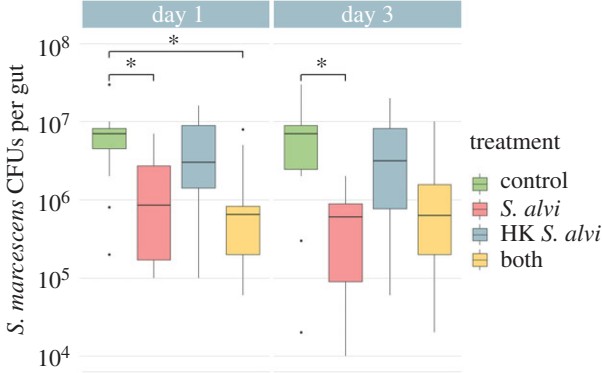

**Figure 4.** Live *S. alvi* aids in the clearance of the pathogen *S. marcescens* while heat-killed *S. alvi* does not. *S. marcescens* CFUs of different treatment groups 1 and 3 days post-inoculation with 5 μl of *S. marcescens* at OD$_{600}$ = 0.5. HK = heat-killed. Bees previously treated with live *S. alvi* and a mixture of live and heat-killed *S. alvi* showed significant reduction of *S. marcescens* CFUs on Day 1 and bees treated with live *S. alvi* continued this trend at Day 3. Bees treated with only heat-killed *S. alvi* showed no significant reduction of *S. marcescens* CFUs compared to controls. Total N = 91 bees from one hive. * = $p < 0.05$. *p*-values obtained using Tukey honest significant difference method. (Online version in colour.)

documented in previous studies, possibly due to our inclusion of bee fat bodies in our assays. Additionally, our results confirm previous studies showing immune priming by heat-killed bacterial cells in insects [37]. In our case, a commensal species provides priming against a pathogenic species, whereas in most studies priming is achieved through exposure to heat-killed pathogenic cells [37].

Comparing bees treated with heat-killed versus live *S. alvi*, the former had the greater upregulation of immune regulatory genes but similar AMP expression (figure 2). However, mRNA levels do not always correspond to protein levels, due to post-transcriptional and post-translational regulatory processes [38]. While little is known about Dorsal and Relish in bees, post-translational regulation of these proteins has been extensively documented in *Drosophila*.

Binding of the inhibitor Cactus prevents the transcription factor Dorsal from translocating to the nucleus [13]. Activation of the Toll pathway causes phosphorylation, ubiquitination and degradation of Cactus [13]. The observed similarities in *dorsal* and *cactus* upregulation may signify that higher expression of *dorsal* is matched by an increase in Cactus inhibitors preventing increases in AMP expression (figures 2*b* and 3*b*).

In the *Drosophila* Imd pathway, Relish activation requires both cleavage and phosphorylation, carried out by other enzymes. Cleavage and phosphorylation are required for translocation to the nucleus and recruitment of RNA polymerase II [14,39]. Therefore, the transcriptional upregulation of *relish* may not lead to a heightened host immune response if these other regulatory processes are hindered. Future studies should investigate how post-transcriptional processes affect expression and should quantify immune gene protein levels in response to these treatments.

Both resistance and tolerance play roles in pathogen defence in animals [40–42]. Resistance reduces pathogen abundance, while tolerance only limits the impact of the pathogen [40–42]. Live and heat-killed *S. alvi* both enhance survival following *S. marcescens* exposure and upregulate immune genes,

but only live *S. alvi* or a mixed treatment aids in pathogen clearance (figures 1 and 4). These results suggest that *S. alvi* plays a role in resistance beyond immune gene upregulation, potentially by direct antagonism or by out-competing invading pathogens for space and resources. However, a protective host response does not imply a fitness benefit.

In *Drosophila*, exposure to *Salmonella typhimurium* or *Listeria monocytogenes* triggers anorexia and altered immune responses, increasing tolerance to the former while decreasing resistance to the latter [43]. Similarly, *Mycobacterium marinum* activates the *Drosophila* immune system, causing the dysregulation of metabolic pathways leading to host wasting and mortality [44]. In bees, lower survivorship following treatment with heat-killed bacteria may indicate a cost for high immune activation (figure 1). Furthermore, lower gene expression in bees treated with live *S. alvi* could reflect reduced investment in costly immune responses made superfluous by the presence of this defensive symbiont.

Bacterial biofilms can prevent the immune system from recognizing bacterial components and clearing an infection [45]. However, we know little about host–biofilm interactions in insects. In tsetse flies, the commensal symbiont *Sodalis glossinidius* requires the outer membrane protein OmpA to form a biofilm and colonize the fly gut [46]. The knockout of *OmpA* causes the clearance of *Sodalis* by the host immune system [46]. Similarly, *S. alvi* requires outer membrane proteins for its colonization of the honeybee, forming a biofilm directly on the gut epithelia of the ileum [6,47]. Like *Sodalis*, this biofilm may prevent targeting of *S. alvi* by immune effectors.

The honeybee gut harbours a diverse community of lytic phages [48,49]. It is likely that *S. alvi* lyses during colonization and growth, however, the dynamics of *S. alvi* and phage interactions remains unclear. Regardless, the biofilm produced by *S. alvi* may prevent lysed bacterial components and host immune receptors from coming into contact. Heat-killed cells cannot form a biofilm, and their components likely spread throughout the gut, interacting with membrane receptors in regions not typically associated with *S. alvi*. These features of heat-killed cells could explain the higher induction by heat-killed *S. alvi* and the mixed treatment for Toll pathway genes.

In the mixed treatment group, bees were fed both live and heat-killed cells simultaneously. Under this simultaneous exposure, heat-killed bacterial components may interact with immune receptors before *S. alvi* can form a biofilm or in places where *S. alvi* does not colonize. Future experiments should compare priority effects between live and heat-killed treatments to determine the impact of *S. alvi* colonization prior to feeding with heat-killed cells and vice versa. Additionally, using *S. alvi* mutants deficient for biofilm production would help to study the biofilm's significance in immune activation [50].

In *Drosophila*, Gram-negative bacteria typically trigger the Imd pathway, while Gram-positive bacteria trigger the Toll pathway [51,52]. Bees given either live or heat-killed *S. alvi* show similar expression patterns for Imd pathway genes, while live *S. alvi* treated bees have lower expression of Toll pathway genes (figure 3). Once live *S. alvi* is mixed with heat-killed cells, Toll pathway gene expression favours that of bees given heat-killed *S. alvi* alone. However, we find a sharp drop in Imd pathway gene expression (figures 3*c* and 5). Therefore, live *S. alvi* appears capable of reducing expression of Imd

**Figure 5.** Proposed effect of live *S. alvi* on host gene expression. Bees treated with heat-killed *S. alvi* (dashed blue ovals) show high upregulation of Toll and Imd pathway components. Addition of live *S. alvi* (filled blue ovals) to heat-killed *S. alvi* triggers a reduction in Imd pathway expression while leaving Toll pathway expression unaffected. Pale yellow denotes *S. alvi*-produced biofilm. Pathway proteins and their localizations inferred from *Drosophila melanogaster*. Toll pathway genes: *toll, cactus, dorsal*. Imd pathway genes: *pgrp-lc, pirk, dredd, relish*. Antimicrobial peptides: *abaecin, apidaecin, hymenoptaecin*. Yellow = lower gene expression relative to control bees. Green = higher gene expression relative to control bees. Grey = no differential gene expression relative to control bees. (Online version in colour.)

pathway genes (figure 5). Knockdown of immune gene levels by RNAi could clarify the roles of Imd and Toll components as well as possibly expose avenues for host–symbiont crosstalk [50].

Coevolution of bees and *S. alvi* over an approximately 80 million-year period has likely resulted in highly specific host–symbiont interaction networks [2]. For example, *S. alvi* is highly resistant to apidaecin, which is present in the gut lumen after inoculation with gut microbes [24]. This antimicrobial resistance may represent a coevolutionary response, allowing the host to manipulate its microbiome while leaving beneficial symbionts unharmed. Limitations in our knowledge of honeybee immune pathways have made investigations of these networks difficult. Future studies will require clarification of molecular mechanisms underlying honeybee immune responses. While this study raises new questions, we show replicable differential responses to the symbiont *S. alvi* by the host immune system. We propose that the gut symbiont *S. alvi* can possibly modulate the host immune system.

Data accessibility. All data used in this study can be found in the electronic supplementary material, dataset provided.

Authors' contributions. All authors assisted in study design and manuscript revisions. Manuscript draft written by R.D.H. Experiments performed by R.D.H. Data analysis performed by R.D.H. and S.P.L.

Competing interests. The authors declare no competing interests.

Funding. This work was supported by National Institutes of Health award R35GM131738 to N.A.M. and by a University of Texas Undergraduate Research Fellowship to R.H.

Acknowledgements. We thank J. Elijah Powell for protocols and experimental assistance, Kim Hammond for laboratory support and bee hive maintenance, and Margaret I. Steele for bacterial OD$_{600}$ enumeration curves as well as assistance with *S. marcescens* clearance assay design.

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
