## [Reviewer comments · Proceedings of the Royal Society B: Biological Sciences]

Review History

RSPB-2020-1184.R0 (Original submission)

Review form: Reviewer 1 (Leonard Foster)

Recommendation

Accept with minor revision (please list in comments)

Scientific importance: Is the manuscript an original and important contribution to its field?

Good

General interest: Is the paper of sufficient general interest?

Good

Quality of the paper: Is the overall quality of the paper suitable?

Excellent

Is the length of the paper justified?

Yes

Should the paper be seen by a specialist statistical reviewer?

No

Do you have any concerns about statistical analyses in this paper? If so, please specify them explicitly in your report.

No

It is a condition of publication that authors make their supporting data, code and materials available - either as supplementary material or hosted in an external repository. Please rate, if applicable, the supporting data on the following criteria.

Is it accessible?

Yes

Is it clear?

Yes

Is it adequate?

Yes

Do you have any ethical concerns with this paper?

No

Comments to the Author

The Gram-negative bacteria *Snodgrassella alvi* is part of the gut flora in honeybees, and its presence is known to be beneficial to the host. Here, the authors examined the effects of live and heat-killed *S. alvi* on host survival against the pathogen *Serratia marcescens*, and their effects on the expression of genes that control and produce several major antimicrobial peptides. It appears that *S. alvi* was able to improve host immunity against the pathogen, though surprisingly the Toll pathway (which is normally upregulated in the presence of Gram-positive bacteria) was more upregulated in the presence of the Gram-negative *S. alvi*, as opposed to the expected upregulation of the Imd pathway. The authors discussed reasons for this occurrence, citing the possibility of host-symbiont crosstalk and co-evolution that may have altered the normal transcription regulation mechanisms of Toll and Imd.

Reviewer's comments:

- As a whole, the work described in this manuscript was done with sound experimental design, producing straightforward results that were well-interpreted by the authors.
- In the discussion, the biochemical pathways discussed by the authors to explain their results appear sound, but it is far too long. This can be far better explained in a diagram, showing their proposed model or hypothesis
- There are some grammatical issues that could be improved to enhance readability. A few examples of problematic sentences:
 - o Line 14 "Honey bee immune genes respond differently to live *S. alvi* compared to heat-killed *S. alvi*, which causes a more extensive immune expression response." It is unclear what "which" is referring to.
 - o Line 24 "This microbial community represents a vital cornerstone of honey bee health, which provide essential environmental and economic services." The clause before and after the comma are not related. Keep them as separate sentences. It is also not clear what "which" is referring to.
 - o Line 21 and Line 24 are redundant about the importance of bee gut microbiome.
- The reasoning for use of *E. coli* as a control needs to be better justified. Why was this species chosen? This isn't a natural bacterium of the bee so how is it a relevant control?

Review form: Reviewer 2

Recommendation

Major revision is needed (please make suggestions in comments)

Scientific importance: Is the manuscript an original and important contribution to its field?

Excellent

General interest: Is the paper of sufficient general interest?

Excellent

Quality of the paper: Is the overall quality of the paper suitable?

Excellent

Is the length of the paper justified?

Yes

Should the paper be seen by a specialist statistical reviewer?

No

Do you have any concerns about statistical analyses in this paper? If so, please specify them explicitly in your report.

No

It is a condition of publication that authors make their supporting data, code and materials available - either as supplementary material or hosted in an external repository. Please rate, if applicable, the supporting data on the following criteria.

Is it accessible?

Yes

Is it clear?

Yes

Is it adequate?

Yes

Do you have any ethical concerns with this paper?

No

Comments to the Author

Overview:

The microbiome of honey bees is known to play many important roles in maintaining the health of individual bees. One mechanism through which bacterial species of the digestive tract support honey bee health is by preventing the colonization of the digestive tract by potentially pathogenic bacteria. Here, the authors explore how *S. alvi*, a symbiont found in the honey bee digestive tract, interferes with colonization of this tissue by the pathogenic bacterium, *S. marcescens*. They show that feeding live or heat-killed *S. alvi* increases honey bee survival after challenge by the pathogenic bacterium, *S. marcescens*. They go on to show that both live and heat-killed *S. alvi* can alter immune gene expression and suggest that this may be the mechanism through which *S. alvi* impacts the outcome of *S. marcescens* infection. These results are quite interesting and represent an important area of research with regards to the bacteria of the honey bee microbiome. They also provide a novel model that may allow for the molecular definition of key systems used by *S. alvi* to modulate honey bee gene expression and biology. Additionally, they provide a model that

may allow for better characterization of the regulation of immune responses in the honey bee. While this study has considerable merit, the work is marked by some significant issues, which make it unsuitable for publication in its present form. A number of more minor issues also exist. These issues must be addressed, resulting in significant improvements, before the manuscript should be considered for publication.

Major Comments:

The authors find that feeding live or heat-killed *S. alvi* increases bee survival after challenge with the pathogenic bacteria *S. marcescens*. The authors do not show that increased survival is through reduced bacteria colonization. In fact, the observed increase in survival could also be mediated through other mechanisms, such as effects that neutralize *S. marcescens* virulence without changing growth or effects on the host that increase tolerance to infection. The authors should show evidence of reduced colonization by *S. marcescens* (either by plating or molecular methods).

Given their findings on the relationship between *S. alvi* and *S. marcescens*, the authors should look at what happens with immune gene expression when bees are challenged with *S. marcescens* +/- live or heat-killed *S. alvi*.

Immune modulation induced by feeding bees with live or heat-killed *S. alvi* may impact the composition of the microbiome, which may then affect subsequent colonization pathogenic bacteria, such as *S. marcescens*. The authors may already know whether *S. alvi* feeding changes other aspects of the microbiome. If so, they should describe these changes. If not, the authors should provide data to show how treatment with live or heat-killed *S. alvi* impacts the species composition of the microbiome.

Minor comments:

Line 35. The authors state that “Microbiome-derived protection may be caused by active symbiont colonization that physically blocks or antagonizes pathogens.” Could the authors describe in more detail other mechanisms through which the microbiome could antagonize pathogens?

Ln 201. The changes in the gene expression of the Imd and Toll pathway components, such as Relish and Dorsal, are interesting and may reflect alterations in pathway architecture over time in response to pathway activation. As the authors state, these components are primarily regulated at the post translational level during signal transmission. While the authors do not need to add additional experimentation to show alterations in the activation or localization of Relish and Dorsal, they should discuss how these proteins are regulated in more detail.

Ln 203. The sentence about wasting is not well integrated into this paragraph. It brings up important points and could likely use its own paragraph that also discusses ideas about resistance vs. tolerance (1), which has been discussed in *Drosophila* model (2).

References:

1. Råberg, L., Sim, D. & Read, A. F. Disentangling genetic variation for resistance and tolerance to infectious diseases in animals. *Science* 318, 812–814 (2007).
2. Ayres, J. S. & Schneider, D. S. The role of anorexia in resistance and tolerance to infections in *Drosophila*. *PLoS Biol* 7, e1000150 (2009).

Decision letter (RSPB-2020-1184.R0)

23-Jun-2020

Dear Mr Horak:

Your manuscript has now been peer reviewed and the reviews have been assessed by an Associate Editor. The reviewers' comments (not including confidential comments to the Editor) and the comments from the Associate Editor are included at the end of this email for your reference. As you will see, the reviewers and the Editors have raised some concerns with your manuscript and we would like to invite you to revise your manuscript to address them.

Research ethics:

Use of animals and field studies:

Please submit a copy of your revised paper within three weeks. If we do not hear from you within this time your manuscript will be rejected. If you are unable to meet this deadline please let us know as soon as possible, as we may be able to grant a short extension.

Best wishes,
Professor Gary Carvalho
mailto: proceedingsb@royalsociety.org

Associate Editor

Board Member: 1

Comments to Author:

The paper has received two reviews, both of which appreciated the value of the study. The first review suggests some fairly minor clarifications to be made in a revision. The second review is requesting more substantive changes, which can be considered in a revision. In particular, the referee is asking for additional information in several areas, including evidence of reduced colonization by *S. marcescens*, gene expression responses in response to *S. marcescens* +/- live or heat-killed *S. alvi*., and additional information about how treatment with live or heat-killed *S. alvi* impacts the species composition of the microbiome.

Reviewer(s)' Comments to Author:

Referee: 1

Comments to the Author(s)

The Gram-negative bacteria *Snodgrassella alvi* is part of the gut flora in honeybees, and its presence is known to be beneficial to the host. Here, the authors examined the effects of live and

heat-killed *S. alvi* on host survival against the pathogen *Serratia marcescens*, and their effects on the expression of genes that control and produce several major antimicrobial peptides. It appears that *S. alvi* was able to improve host immunity against the pathogen, though surprisingly the Toll pathway (which is normally upregulated in the presence of Gram-positive bacteria) was more upregulated in the presence of the Gram-negative *S. alvi*, as opposed to the expected upregulation of the Imd pathway. The authors discussed reasons for this occurrence, citing the possibility of host-symbiont crosstalk and co-evolution that may have altered the normal transcription regulation mechanisms of Toll and Imd.

Reviewer's comments:

- As a whole, the work described in this manuscript was done with sound experimental design, producing straightforward results that were well-interpreted by the authors.
- In the discussion, the biochemical pathways discussed by the authors to explain their results appear sound, but it is far too long. This can be far better explained in a diagram, showing their proposed model or hypothesis
- There are some grammatical issues that could be improved to enhance readability. A few examples of problematic sentences:
 - o Line 14 "Honey bee immune genes respond differently to live *S. alvi* compared to heat-killed *S. alvi*, which causes a more extensive immune expression response." It is unclear what "which" is referring to.
 - o Line 24 "This microbial community represents a vital cornerstone of honey bee health, which provide essential environmental and economic services." The clause before and after the comma are not related. Keep them as separate sentences. It is also not clear what "which" is referring to.
 - o Line 21 and Line 24 are redundant about the importance of bee gut microbiome.
- The reasoning for use of *E. coli* as a control needs to be better justified. Why was this species chosen? This isn't a natural bacterium of the bee so how is it a relevant control?

Referee: 2

Comments to the Author(s)

Overview:

The microbiome of honey bees is known to play many important roles in maintaining the health of individual bees. One mechanism through which bacterial species of the digestive tract support honey bee health is by preventing the colonization of the digestive tract by potentially pathogenic bacteria. Here, the authors explore how *S. alvi*, a symbiont found in the honey bee digestive tract, interferes with colonization of this tissue by the pathogenic bacterium, *S. marcescens*. They show that feeding live or heat-killed *S. alvi* increases honey bee survival after challenge by the pathogenic bacterium, *S. marcescens*. They go on to show that both live and heat-killed *S. alvi* can alter immune gene expression and suggest that this may be the mechanism through which *S. alvi* impacts the outcome of *S. marcescens* infection. These results are quite interesting and represent an important area of research with regards to the bacteria of the honey bee microbiome. They also provide a novel model that may allow for the molecular definition of key systems used by *S. alvi* to modulate honey bee gene expression and biology. Additionally, they provide a model that may allow for better characterization of the regulation of immune responses in the honey bee. While this study has considerable merit, the work is marked by some significant issues, which make it unsuitable for publication in its present form. A number of more minor issues also exist. These issues must be addressed, resulting in significant improvements, before the manuscript should be considered for publication.

Major Comments:

The authors find that feeding live or heat-killed *S. alvi* increases bee survival after challenge with the pathogenic bacteria *S. marcescens*. The authors do not show that increased survival is through reduced bacteria colonization. In fact, the observed increase in survival could also be

mediated through other mechanisms, such as effects that neutralize *S. marcescens* virulence without changing growth or effects on the host that increase tolerance to infection. The authors should show evidence of reduced colonization by *S. marcescens* (either by plating or molecular methods).

Given their findings on the relationship between *S. alvi* and *S. marcescens*, the authors should look at what happens with immune gene expression when bees are challenged with *S. marcescens* +/- live or heat-killed *S. alvi*.

Immune modulation induced by feeding bees with live or heat-killed *S. alvi* may impact the composition of the microbiome, which may then affect subsequent colonization pathogenic bacteria, such as *S. marcescens*. The authors may already know whether *S. alvi* feeding changes other aspects of the microbiome. If so, they should describe these changes. If not, the authors should provide data to show how treatment with live or heat-killed *S. alvi* impacts the species composition of the microbiome.

Minor comments:

Line 35. The authors state that "Microbiome-derived protection may be caused by active symbiont colonization that physically blocks or antagonizes pathogens." Could the authors describe in more detail other mechanisms through which the microbiome could antagonize pathogens?

Ln 201. The changes in the gene expression of the Imd and Toll pathway components, such as Relish and Dorsal, are interesting and may reflect alterations in pathway architecture over time in response to pathway activation. As the authors state, these components are primarily regulated at the post translational level during signal transmission. While the authors do not need to add additional experimentation to show alterations in the activation or localization of Relish and Dorsal, they should discuss how these proteins are regulated in more detail.

Ln 203. The sentence about wasting is not well integrated into this paragraph. It brings up important points and could likely use its own paragraph that also discusses ideas about resistance vs. tolerance (1), which has been discussed in *Drosophila* model (2).

References:

1. Råberg, L., Sim, D. & Read, A. F. Disentangling genetic variation for resistance and tolerance to infectious diseases in animals. *Science* 318, 812–814 (2007).
2. Ayres, J. S. & Schneider, D. S. The role of anorexia in resistance and tolerance to infections in *Drosophila*. *PLoS Biol* 7, e1000150 (2009).

Author's Response to Decision Letter for (RSPB-2020-1184.R0)

See Appendix A.

Decision letter (RSPB-2020-1184.R1)

04-Aug-2020

Dear Mr Horak

I am pleased to inform you that your manuscript entitled "Symbionts shape host innate immunity in honey bees" has been accepted for publication in Proceedings B.

Open Access

Your article has been estimated as being 9 pages long. Our Production Office will be able to confirm the exact length at proof stage.

Paper charges

Sincerely,

Professor Gary Carvalho

Associate Editor:

Board Member

Comments to Author:

We have reassessed your manuscript and were pleased to see the thorough revisions, inclusion of results of an additional experiment, and new schematic figure (Figure 5). We find it much improved and suitable for publication as is. Thank you for submitting your interesting work to PRSB.

Response to Reviewers

General Response

We would like to thank both reviewers for reading our manuscript and providing constructive comments. Our revisions include an additional experiment, showing that live *S. alvi* is capable of clearing *S. marcescens* pathogens from the gut (Figure 4). These results aided the inclusion of a discussion on resistance versus tolerance and the potential presence of these mechanisms in the honey bee gut system. We also addressed more minor concerns, providing further justification for *E. coli* as a test condition, removal of grammatical issues and extraneous information, inclusion of a discussion on Relish and Dorsal post-translational regulation, and the addition of a graphical illustration of the proposed effects of live *S. alvi* on honey bee immune responses (Figure 5).

Specific Responses to Reviewer 1

Reviewer 1

- As a whole, the work described in this manuscript was done with sound experimental design, producing straightforward results that were well-interpreted by the authors.
- In the discussion, the biochemical pathways discussed by the authors to explain their results appear sound, but it is far too long. This can be far better explained in a diagram, showing their proposed model or hypothesis

We have condensed the parts of the discussion pertaining to explanation of results (Lines 218-239, 297-314). Additionally, we have added a graphical illustration of the proposed effects of live *S. alvi* on the honey bee immune system (Figure 5).

- There are some grammatical issues that could be improved to enhance readability. A few examples of problematic sentences:
 - o Line 14 “Honey bee immune genes respond differently to live *S. alvi* 14 compared to heat-killed *S. alvi*, which causes a more extensive immune expression response.” It is unclear what “which” is referring to.
 - o Line 24 “This microbial community represents a vital cornerstone of honey bee health, which provide essential environmental and economic services.” The clause before and after the comma are not related. Keep them as separate sentences. It is also not clear what “which” is referring to.
 - o Line 21 and Line 24 are redundant about the importance of bee gut microbiome.

We corrected all these issues. Additionally, we corrected similar issues throughout the text, reducing redundancy and removing all unclear antecedents.

- The reasoning for use of *E. coli* as a control needs to be better justified. Why was this species chosen? This isn't a natural bacterium of the bee so how is it a relevant control?

We agree that our previous justification left it unclear as to the purpose of the *E. coli* treatment. We have expanded our justification for the use of *E. coli* as a test condition (Lines 158-163). As both *E. coli* and *S. alvi* are Gram-negative, but only *S. alvi* is a symbiont, we believed *E. coli* could shed light on whether any expression responses were specific to symbionts, or general to Gram-negative bacteria.

Specific Responses to Reviewer 2

Reviewer 2

Major Comments:

The authors find that feeding live or heat-killed *S. alvi* increases bee survival after challenge with the pathogenic bacteria *S. marcescens*. The authors do not show that increased survival is through reduced bacteria colonization. In fact, the observed increase in survival could also be mediated through other mechanisms, such as effects that neutralize *S. marcescens* virulence without changing growth or effects on the host that increase tolerance to infection. The authors should show evidence of reduced colonization by *S. marcescens* (either by plating or molecular methods).

This is an important question, and we agree that determining whether *S. alvi* aids in pathogen reduction is important for our conclusions. We have now performed a new experiment, treating microbiota-free bees with buffer only, live *S. alvi*, heat-killed *S. alvi*, or a mixture of the two and then challenging them with *S. marcescens*. We then spot plated dilutions from each gut to determine *S. marcescens* CFUs 1 day and 3 days post *S. marcescens* inoculation. We found that bees treated with live *S. alvi* and the mixture of live and dead *S. alvi* had fewer *S. marcescens* CFUs, but treatment with heat-killed *S. alvi* had no effect (Lines 124-128, 204-215, 252-267, Figure 4).

Given their findings on the relationship between *S. alvi* and *S. marcescens*, the authors should look at what happens with immune gene expression when bees are challenged with *S. marcescens* +/- live or heat-killed *S. alvi*.

While this is an interesting question, we believe the results of such an experiment are beyond the scope of this paper. We sought to look at this system from the perspective of the symbiont *S. alvi* and its role in immune regulation in its honey bee host. Therefore, we believe this experiment is better suited for a study more focused around the pathogen *S. marcescens*. Regardless, we thank the reviewer for suggesting this interesting potential future experiment.

Immune modulation induced by feeding bees with live or heat-killed *S. alvi* may impact the composition of the microbiome, which may then affect subsequent colonization pathogenic bacteria, such as *S. marcescens*. The authors may already know whether *S. alvi* feeding changes other aspects of the microbiome. If so, they should describe these changes. If not, the authors

should provide data to show how treatment with live or heat-killed *S. alvi* impacts the species composition of the microbiome.

All bees used in these experiments were removed from brood cells as pupae and allowed to develop in a sterile environment (Lines 86-87). Bees collected at this stage lack any core microbiota and are microbiota-free. In these experiments, there should be no additional interactions between *S. alvi* or *S. marcescens* with other core gut microbes. To clarify this point we added a sentence explicitly stating this and added a citation to a study describing the lack of microbes in honey bee pupae (Lines 86-87, [26])

26. Powell JE, Martinson VG, Urban-Mead K, Moran NA. 2014 Routes of acquisition of the gut microbiota of the honey bee *Apis mellifera*. *Applied and Environmental Microbiology* 80, 7378–7387. (doi:10.1128/aem.01861-14)

Minor comments:

Line 35. The authors state that “Microbiome-derived protection may be caused by active symbiont colonization that physically blocks or antagonizes pathogens.” Could the authors describe in more detail other mechanisms through which the microbiome could antagonize pathogens?

We now describe several possibilities including physical blockage to epithelial cells, competition for space and resources, type VI secretion system toxins, and alterations to the gut lumen environment by bacterial metabolism. We also cite evidence for these mechanisms in *S. alvi* and the honey bee system (Lines 39-43, [6,10-12])

6. Martinson VG, Moy J, Moran NA. 2012 Establishment of characteristic gut bacteria during development of the honeybee worker. *Applied and Environmental Microbiology* 78, 2830–2840. (doi:10.1128/aem.07810-11)

10. Steele MI, Kwong WK, Whiteley M, Moran NA. 2017 Diversification of type VI secretion system toxins reveals ancient antagonism among bee gut microbes. *mBio* 8. (doi:10.1128/mbio.01630-17)

11. Pickard JM, Zeng MY, Caruso R, Núñez G. 2017 Gut microbiota: Role in pathogen colonization, immune responses, and inflammatory disease. *Immunological Reviews* 279, 70–89. (doi:10.1111/imr.12567)

12. Zheng H, Powell JE, Steele MI, Dietrich C, Moran NA. 2017 Honeybee gut microbiota promotes host weight gain via bacterial metabolism and hormonal signaling. *Proc Natl Acad Sci USA* 114, 4775–4780. (doi:10.1073/pnas.1701819114)

Ln 201. The changes in the gene expression of the Imd and Toll pathway components, such as Relish and Dorsal, are interesting and may reflect alterations in pathway architecture over time in response to pathway activation. As the authors state, these components are primarily

regulated at the post translational level during signal transmission. While the authors do not need to add additional experimentation to show alterations in the activation or localization of Relish and Dorsal, they should discuss how these proteins are regulated in more detail.

We expanded our discussion of post-translational regulation of Relish and Dorsal (Lines 237-249, [39]). These regulatory mechanisms come from *Drosophila* studies as very little is known about the molecular biology of the Imd and Toll pathways in honey bees.

39. Erturk-Hasdemir D et al. 2009 Two roles for the *Drosophila* IKK complex in the activation of Relish and the induction of antimicrobial peptide genes. *Proceedings of the National Academy of Sciences* 106, 9779–9784. (doi:10.1073/pnas.0812022106)

Ln 203. The sentence about wasting is not well integrated into this paragraph. It brings up important points and could likely use its own paragraph that also discusses ideas about resistance vs. tolerance (1), which has been discussed in *Drosophila* model (2).

References:

1. Råberg, L., Sim, D. & Read, A. F. Disentangling genetic variation for resistance and tolerance to infectious diseases in animals. *Science* 318, 812–814 (2007).
2. Ayres, J. S. & Schneider, D. S. The role of anorexia in resistance and tolerance to infections in *Drosophila*. *PLoS Biol* 7, e1000150 (2009).

We appreciate this point and the references which aided in our own education about these mechanisms. We have expanded the discussion to include these points, integrate the sentence on wasting in *Drosophila* more seamlessly, and include our new results showing that *S. alvi* provide resistance to *S. marcescens* (Lines 252-267, [40-44]). We propose that both tolerance and resistance exist in this system, with tolerance conferred by live or dead *S. alvi*, triggering an immune response which improves survivorship but does not lower *S. marcescens* growth, and resistance conferred by live *S. alvi*, likely a microbe-microbe interaction of antagonism or competition.

40. Raberg L, Sim D, Read AF. 2007 Disentangling genetic variation for resistance and tolerance to infectious diseases in animals. *Science* 318, 812–814. (doi:10.1126/science.1148526)

41. Read AF, Graham AL, Råberg L. 2008 Animal defenses against infectious agents: Is damage control more important than pathogen control. *PLoS Biology* 6, e1000004. (doi:10.1371/journal.pbio.1000004)

42. Schneider DS, Ayres JS. 2008 Two ways to survive infection: what resistance and tolerance can teach us about treating infectious diseases. *Nature Reviews Immunology* 8, 889–895. (doi:10.1038/nri2432)

43. Ayres JS, Schneider DS. 2009 The role of anorexia in resistance and tolerance to infections in *Drosophila*. PLoS Biology 7, e1000150. (doi:10.1371/journal.pbio.1000150)
44. Dionne MS, Pham LN, Shirasu-Hiza M, Schneider DS. 2006 Akt and foxo dysregulation contribute to infection-induced wasting in *Drosophila*. Current Biology 16, 1977–1985. (doi:10.1016/j.cub.2006.08.052)